# Ordered Neurons: Integrating Tree Structures into Recurrent Neural Networks

**Yikang Shen**[*]
Mila/Université de Montréal and Microsoft Research
Montréal, Canada

**Shawn Tan**[*]
Mila/Université de Montréal
Montréal, Canada

**Alessandro Sordoni**
Microsoft Research
Montréal, Canada

**Aaron Courville**
Mila/Université de Montréal
Montréal, Canada

## Abstract

Natural language is hierarchically structured: smaller units (e.g., phrases) are nested within larger units (e.g., clauses). When a larger constituent ends, all of the smaller constituents that are nested within it must also be closed. While the standard LSTM architecture allows different neurons to track information at different time scales, it does not have an explicit bias towards modeling a hierarchy of constituents. This paper proposes to add such an inductive bias by *ordering* the neurons; a vector of master input and forget gates ensures that when a given neuron is updated, all the neurons that follow it in the ordering are also updated. Our novel recurrent architecture, *ordered neurons* LSTM (ON-LSTM), achieves good performance on four different tasks: language modeling, unsupervised parsing, targeted syntactic evaluation, and logical inference[1].

## 1 Introduction

Natural language has a sequential overt form as spoken and written, but the underlying structure of language is not strictly sequential. This structure is usually tree-like. Linguists agree on a set of rules, or *syntax*, that determine this structure (Chomsky, 1956; 1965; Sandra & Taft, 2014) and dictate how single words compose to form meaningful larger units, also called "constituents" (Koopman et al., 2013). The human brain can also implicitly acquire the latent structure of language (Dehaene et al., 2015): during language acquisition, children are not given annotated parse trees. This observation brings more interest in latent structure induction with artificial neural network approaches, which are inspired by information processing and communication patterns in biological nervous systems. From a practical point of view, integrating a tree structure into a neural network language model may be important for multiple reasons:

(i) to obtain a hierarchical representation with increasing levels of abstraction, a key feature of deep neural networks (Bengio et al., 2009; LeCun et al., 2015; Schmidhuber, 2015);

(ii) to model the compositional effects of language (Koopman et al., 2013; Socher et al., 2013) and help with the long-term dependency problem (Bengio et al., 2009; Tai et al., 2015) by providing shortcuts for gradient backpropagation (Chung et al., 2016);

(iii) to improve generalization via a better inductive bias and at the same time potentially reducing the need of a large amount of training data.

The study of deep neural network techniques that can infer and use tree structures to form better representations of natural language sentences has received a great deal of attention in recent

---

[*]Equal contribution. `{yi-kang.shen,jing.shan.shawn.tan}@umontreal.ca`.

[1]The code can be found at `https://github.com/yikangshen/Ordered-Neurons`.

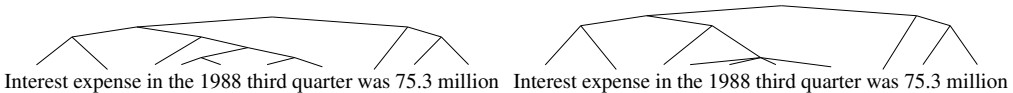

Figure 1: Binary parse tree inferred by our model (left) and its corresponding ground-truth (right).

years (Bowman et al., 2016; Yogatama et al., 2016; Shen et al., 2017; Jacob et al., 2018; Choi et al., 2018; Williams et al., 2018; Shi et al., 2018).

Given a sentence, one straightforward way of predicting the corresponding latent tree structure is through a supervised syntactic parser. Trees produced by these parsers have been used to guide the composition of word semantics into sentence semantics (Socher et al., 2013; Bowman et al., 2015), or even to help next word prediction given previous words (Wu et al., 2017). However, supervised parsers are limiting for several reasons: i) few languages have comprehensive annotated data for supervised parser training; ii) in some domains, syntax rules tend to be broken (e.g. in tweets); and iii) languages change over time with use, so syntax rules may evolve.

On the other hand, *grammar induction*, defined as the task of learning the syntactic structure from raw corpora without access to expert-labeled data, remains an open problem. Many such recent attempts suffer from inducing a trivial structure (e.g., a left-branching or right-branching tree (Williams et al., 2018)), or encounter difficulties in training caused by learning branching policies with Reinforcement Learning (RL) (Yogatama et al., 2016). Furthermore, some methods are relatively complex to implement and train, like the PRPN model proposed in Shen et al. (2017).

Recurrent neural networks (RNNs) have proven highly effective at the task of language modeling (Merity et al., 2017; Melis et al., 2017). RNNs explicitly impose a chain structure on the data. This assumption may seem at odds with the latent non-sequential structure of language and may pose several difficulties for the processing of natural language data with deep learning methods, giving rise to problems such as capturing long-term dependencies (Bengio et al., 2009), achieving good generalization (Bowman et al., 2015), handling negation (Socher et al., 2013), etc. Meanwhile, some evidence exists that LSTMs with sufficient capacity potentially implement syntactic processing mechanisms by encoding the tree structure implicitly, as shown by Gulordava et al. (2018); Kuncoro et al. (2018) and very recently by Lakretz et al. (2019). We believe that the following question remains: Can better models of language be obtained by architectures equipped with an inductive bias towards learning such latent tree structures?

In this work, we introduce *ordered neurons*, a new inductive bias for recurrent neural networks. This inductive bias promotes differentiation of the life cycle of information stored inside each neuron: high-ranking neurons will store long-term information which is kept for a large number of steps, while low-ranking neurons will store short-term information that can be rapidly forgotten. To avoid a strict division between high-ranking and low-ranking neurons, we propose a new activation function, the cumulative softmax, or cumax(), to actively allocate neurons to store long/short-term information. We use the cumax() function to produce a vector of master input and forget gates ensuring that when a given neuron is updated (erased), all of the neurons that follow it in the ordering are also updated (erased). Based on the cumax() and the LSTM architecture, we have designed a new model, ON-LSTM, that is biased towards performing tree-like composition operations. Our model achieves good performance on four tasks: language modeling, unsupervised constituency parsing, targeted syntactic evaluation (Marvin & Linzen, 2018) and logical inference (Bowman et al., 2015). The result on unsupervised constituency parsing suggests that the proposed inductive bias aligns with the syntax principles proposed by human experts better than previously proposed models. The experiments also show that ON-LSTM performs better than standard LSTM models in tasks requiring capturing long-term dependencies and achieves better generalization to longer sequences.

## 2 RELATED WORK

There has been prior work leveraging tree structures for natural language tasks in the literature. Socher et al. (2010); Alvarez-Melis & Jaakkola (2016); Zhou et al. (2017); Zhang et al. (2015) use supervised learning on expert-labeled treebanks for predicting parse trees. Socher et al. (2013) and Tai et al. (2015) explicitly model the tree-structure using parsing information from an external

parser. Later, Bowman et al. (2016) exploited guidance from a supervised parser (Klein & Manning, 2003) in order to train a stack-augmented neural network.

Theoretically, RNNs and LSTMs can model data produced by context-free grammars and context-sensitive grammars (Gers & Schmidhuber, 2001). However, recent results suggest that introducing structure information into LSTMs is beneficial. Kuncoro et al. (2018) showed that RNNGs (Dyer et al., 2016), which have an explicit bias to model the syntactic structures, outperform LSTMs on the subject-verb agreement task (Linzen et al., 2016). In our paper, we run a more extensive suite of grammatical tests recently provided by Marvin & Linzen (2018). Bowman et al. (2014; 2015) also demonstrate that tree-structured models are more effective for downstream tasks whose data was generated by recursive programs. Interestingly, Shi et al. (2018) suggests that while the prescribed grammar tree may not be ideal, some sort of hierarchical structure, perhaps task dependent, might help. However, the problem of efficiently inferring such structures from observed data remains an open question.

The task of learning the underlying grammar from data is known as *grammar induction* (Chen, 1995; Cohen et al., 2011). Early work incorporated syntactic structure in the context of language modeling (Roark, 2001; Charniak, 2001; Chelba & Jelinek, 2000). More recently, there have been attempts at incorporating some structure for downstream tasks using neural models (Grefenstette et al., 2015; Sun et al., 2017; Joulin & Mikolov, 2015). Generally, these works augment a main recurrent model with a stack and focus on solving algorithmic tasks. Yogatama et al. (2018) focus on language modeling and syntactic evaluation tasks (Linzen et al., 2016) but they do not show the extent to which the structure learnt by the model align with gold-standard parse trees. Shen et al. (2017) introduced the Parsing-Reading-Predict Networks (PRPN) model, which attempts to perform parsing by solving a language modeling task. The model uses self-attention to compose previous states, where the range of attention is controlled by a learnt "syntactic distance". The authors show that this value corresponds to the depth of the parse tree. However, the added complexity in using the PRPN model makes it unwieldy in practice.

Another possible solution is to develop models with varying time-scales of recurrence as a way of capturing this hierarchy. El Hihi & Bengio (1996); Schmidhuber (1991); Lin et al. (1998) describe models that capture hierarchies at pre-determined time-scales. More recently, Koutnik et al. (2014) proposed Clockwork RNN, which segments the hidden state of a RNN by updating at different time-scales. These approaches typically make a strong assumption about the regularity of the hierarchy involved in modelling the data. Chung et al. (2016) proposed a method that, unlike the Clockwork RNN, would learn a multi-scale hierarchical recurrence. However, the model still has a pre-determined depth to the hierarchy, depending on the number of layers. Our work is more closely related to Rippel et al. (2014), which propose to induce a hierarchy in the representation units by applying "nested" dropout masks: units are not dropped independently at random but whenever a unit is dropped, all the units that follow in the ordering are also dropped. Our work can be seen as a soft relaxation of the dropout by means of the proposed $\mathrm{cumax}()$ activation. Moreover, we propose to condition the update masks on the particular input and apply our overall model to sequential data. Therefore, our model can adapt the structure to the observed data, while both Clockwork RNN and nested dropout impose a predefined hierarchy to hidden representations.

## 3 ORDERED NEURONS

Given a sequence of tokens $S = (x_1, \ldots, x_T)$ and its corresponding constituency tree (Figure 2(a)), our goal is to infer the unobserved tree structure while processing the observed sequence, i.e. while computing the hidden state $h_t$ for each time step $t$. At each time step, $h_t$ would ideally contain a information about all the nodes on the path between the current leaf node $x_t$ and the root S. In Figure 2(c), we illustrate how $h_t$ would contain information about all the constituents that include the current token $x_t$ even if those are only partially observed. This intuition suggests that each node in the tree can be represented by a set of neurons in the hidden states. However, while the dimensionality of the hidden state is fixed in advance, the length of the path connecting the leaf to the root of the tree may be different across different time steps and sentences. Therefore, a desiderata for the model is to dynamically reallocate the dimensions of the hidden state to each node.

Given these requirements, we introduce *ordered neurons*, an inductive bias that forces neurons to represent information at different time-scales. In our model, high-ranking neurons contain long-term

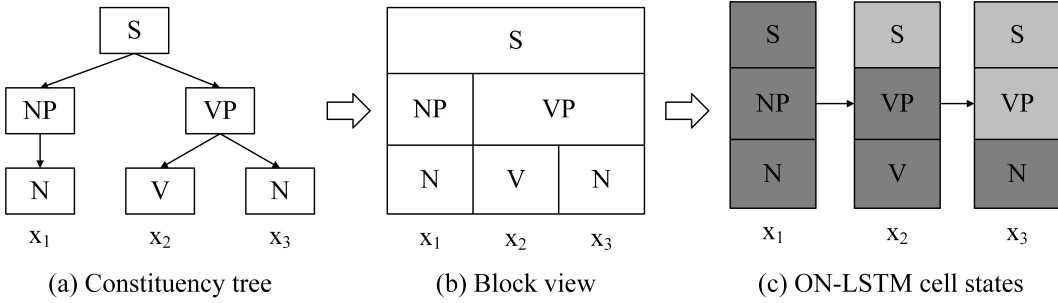

|              |              |              |
| :----------: | :----------: | :----------: |
| (a) Constituency tree | (b) Block view | (c) ON-LSTM cell states |

Figure 2: Correspondences between a constituency parse tree and the hidden states of the proposed ON-LSTM. A sequence of tokens $S = (x_1, x_2, x_3)$ and its corresponding constituency tree are illustrated in (a). We provide a block view of the tree structure in (b), where both S and VP nodes span more than one time step. The representation for high-ranking nodes should be relatively consistent across multiple time steps. (c) Visualization of the update frequency of groups of hidden state neurons. At each time step, given the input word, dark grey blocks are completely updated while light grey blocks are partially updated. The three groups of neurons have different update frequencies. Topmost groups update less frequently while lower groups are more frequently updated.

or global information that will last anywhere from several time steps to the entire sentence, representing nodes near the root of the tree. Low-ranking neurons encode short-term or local information that only last one or a few time steps, representing smaller constituents, as shown in Figure 2(b). The differentiation between high-ranking and low-ranking neurons is learnt in a completely data-driven fashion by controlling the update frequency of single neurons: to erase (or update) high-ranking neurons, the model should first erase (or update) all lower-ranking neurons. In other words, some neurons always update more (or less) frequently than the others, and that order is pre-determined as part of the model architecture.

## 4    ON-LSTM

In this section, we present a new RNN unit, ON-LSTM ("*ordered neurons* LSTM"). The new model uses an architecture similar to the standard LSTM, reported below:

$$f_t = \sigma(W_f x_t + U_f h_{t-1} + b_f) \tag{1}$$
$$i_t = \sigma(W_i x_t + U_i h_{t-1} + b_i) \tag{2}$$
$$o_t = \sigma(W_o x_t + U_o h_{t-1} + b_o) \tag{3}$$
$$\hat{c}_t = \tanh(W_c x_t + U_c h_{t-1} + b_c) \tag{4}$$
$$h_t = o_t \circ \tanh(c_t) \tag{5}$$

The difference with the LSTM is that we replace the update function for the cell state $c_t$ with a new function that will be explained in the following sections. The forget gates $f_t$ and input gates $i_t$ are used to control the erasing and writing operation on cell states $c_t$, as before. Since the gates in the LSTM act independently on each neuron, it may be difficult in general to discern a hierarchy of information between the neurons. To this end, we propose to make the gate for each neuron dependent on the others by enforcing the order in which neurons should be updated.

### 4.1    ACTIVATION FUNCTION: cumax()

To enforce an order to the update frequency, we introduce a new activation function:

$$\hat{g} = \mathrm{cumax}(\dots) = \mathrm{cumsum}(\mathrm{softmax}(\dots)), \tag{6}$$

where cumsum denotes the cumulative sum. We will show that the vector $\hat{g}$ can be seen as the expectation of a binary gate $g = (0, ..., 0, 1, ..., 1)$. This binary gate splits the cell state into two segments: the 0-segment and the 1-segment. Thus, the model can apply different update rules on the two segments to differentiate long/short-term information. Denote by $d$ a categorical random

variable representing the index for the first 1 in $g$:

$$p(d) = \text{softmax}(\ldots) \tag{7}$$

The variable $d$ represents the split point between the two segments. We can compute the probability of the $k$-th value in $g$ being 1 by evaluating the probability of the disjunction of any of the values before the $k$-th being the split point, that is $d \leq k = (d = 0) \vee (d = 1) \vee \cdots \vee (d = k)$. Since the categories are mutually exclusive, we can do this by computing the cumulative distribution function:

$$p(g_k = 1) = p(d \leq k) = \sum_{i \leq k} p(d = i) \tag{8}$$

Ideally, $g$ should take the form of a discrete variable. Unfortunately, computing gradients when a discrete variable is included in the computation graph is not trivial (Schulman et al., 2015), so in practice we use a continuous relaxation by computing the quantity $p(d \leq k)$, obtained by taking a cumulative sum of the softmax. As $g_k$ is binary, this is equivalent to computing $\mathbb{E}[g_k]$. Hence, $\hat{g} = \mathbb{E}[g]$.

## 4.2 STRUCTURED GATING MECHANISM

Based on the $\text{cumax}()$ function, we introduce a master forget gate $\tilde{f}_t$ and a master input gate $\tilde{i}_t$:

$$\tilde{f}_t = \text{cumax}(W_{\tilde{f}} x_t + U_{\tilde{f}} h_{t-1} + b_{\tilde{f}}) \tag{9}$$

$$\tilde{i}_t = 1 - \text{cumax}(W_{\tilde{i}} x_t + U_{\tilde{i}} h_{t-1} + b_{\tilde{i}}) \tag{10}$$

Following the properties of the $\text{cumax}()$ activation, the values in the master forget gate are monotonically increasing from 0 to 1, and those in the master input gate are monotonically decreasing from 1 to 0. These gates serve as high-level control for the update operations of cell states. Using the master gates, we define a new update rule:

$$\omega_t = \tilde{f}_t \circ \tilde{i}_t \tag{11}$$

$$\hat{f}_t = f_t \circ \omega_t + (\tilde{f}_t - \omega_t) = \tilde{f}_t \circ (f_t \circ \tilde{i}_t + 1 - \tilde{i}_t) \tag{12}$$

$$\hat{i}_t = i_t \circ \omega_t + (\tilde{i}_t - \omega_t) = \tilde{i}_t \circ (i_t \circ \tilde{f}_t + 1 - \tilde{f}_t) \tag{13}$$

$$c_t = \hat{f}_t \circ c_{t-1} + \hat{i}_t \circ \hat{c}_t \tag{14}$$

In order to explain the intuition behind the new update rule, we assume that the master gates are binary:

- The master forget gate $\tilde{f}_t$ controls the erasing behavior of the model. Suppose $\tilde{f}_t = (0, \ldots, 0, 1, \ldots, 1)$ and the split point is $d_t^f$. Given the Eq. (12) and (14), the information stored in the first $d_t^f$ neurons of the previous cell state $c_{t-1}$ will be completely erased. In a parse tree (e.g. Figure 2(a)), this operation is akin to closing previous constituents. A large number of zeroed neurons, i.e. a large $d_t^f$, represents the end of a high-level constituent in the parse tree, as most of the information in the state will be discarded. Conversely, a small $d_t^f$ represents the end of a low-level constituent as high-level information is kept for further processing.

- The master input gate $\tilde{i}_t$ is meant to control the writing mechanism of the model. Assume that $\tilde{i}_t = (1, \ldots, 1, 0, \ldots, 0)$ and the split point is $d_t^i$. Given Eq. (13) and (14), a large $d_t^i$ means that the current input $x_t$ contains long-term information that needs to be preserved for several time steps. Conversely, a small $d_t^i$ means that the current input $x_t$ just provides local information that could be erased by $\tilde{f}_t$ in the next few time steps.

- The product of the two master gates $\omega_t$ represents the overlap of $\tilde{f}_t$ and $\tilde{i}_t$. Whenever an overlap exists ($\exists k, \omega_{tk} > 0$), the corresponding segment of neurons encodes the incomplete constituents that contain some previous words and the current input word $x_t$. Since these constituents are incomplete, we want to update the information inside the respective blocks. The segment is further controlled by the $f_t$ and $i_t$ in the standard LSTM model to enable more fine-grained operations within blocks. For example, in Figure 2, the word $x_3$ is nested

| Model | Parameters | Validation | Test |
|---|---|---|---|
| Zaremba et al. (2014) - LSTM (large) | 66M | 82.2 | 78.4 |
| Gal & Ghahramani (2016) - Variational LSTM (large, MC) | 66M | – | 73.4 |
| Kim et al. (2016) - CharCNN | 19M | – | 78.9 |
| Merity et al. (2016) - Pointer Sentinel-LSTM | 21M | 72.4 | 70.9 |
| Grave et al. (2016) - LSTM | – | – | 82.3 |
| Grave et al. (2016) - LSTM + continuous cache pointer | – | – | 72.1 |
| Inan et al. (2016) - Variational LSTM (tied) + augmented loss | 51M | 71.1 | 68.5 |
| Zilly et al. (2016) - Variational RHN (tied) | 23M | 67.9 | 65.4 |
| Zoph & Le (2016) - NAS Cell (tied) | 54M | – | 62.4 |
| Shen et al. (2017) - PRPN-LM | – | – | 62.0 |
| Melis et al. (2017) - 4-layer skip connection LSTM (tied) | 24M | 60.9 | 58.3 |
| Merity et al. (2017) - AWD-LSTM - 3-layer LSTM (tied) | 24M | 60.0 | 57.3 |
| **ON-LSTM** - 3-layer (tied) | 25M | $\mathbf{58.29 \pm 0.10}$ | $\mathbf{56.17 \pm 0.12}$ |
| Yang et al. (2017) - AWD-LSTM-MoS* | 22M | 56.5 | 54.4 |

Table 1: Single model perplexity on validation and test sets for the Penn Treebank language modeling task. Models labelled *tied* use weight tying on the embedding and softmax weights (Inan et al., 2016; Press & Wolf, 2017). Models labelled * focus on improving the softmax component of RNN language model. Their contribution is orthogonal to ours.

into the constituents S and VP. At this time step, the overlap gray blocks would represent these constituents, such that $\tilde{f}_t$ and $\tilde{i}_t$ can decide whether to reset or update each individual neurons in these blocks.

As the master gates only focus on coarse-grained control, modeling them with the same dimensions as the hidden states is computationally expensive and unnecessary. In practice, we set $\tilde{f}_t$ and $\tilde{i}_t$ to be $D_m = \frac{D}{C}$ dimensional vectors, where $D$ is the dimension of hidden state, and $C$ is a chunk size factor. We repeat each dimension $C$ times, before the element-wise multiplication with $f_t$ and $i_t$. The downsizing significantly reduces the number of extra parameters that we need to add to the LSTM. Therefore, every neuron within each $C$-sized chunk shares the same master gates.

## 5 EXPERIMENTS

We evaluate the proposed model on four tasks: language modeling, unsupervised constituency parsing, targeted syntactic evaluation (Marvin & Linzen, 2018), and logical inference (Bowman et al., 2015).

### 5.1 LANGUAGE MODELING

Word-level language modeling is a macroscopic evaluation of the model's ability to deal with various linguistic phenomena (e.g. co-occurence, syntactic structure, verb-subject agreement, etc). We evaluate our model by measuring perplexity on the Penn TreeBank (PTB) (Marcus et al., 1993; Mikolov, 2012) task.

For fair comparison, we closely follow the model hyper-parameters, regularization and optimization techniques introduced in AWD-LSTM (Merity et al., 2017). Our model uses a three-layer ON-LSTM model with 1150 units in the hidden layer and an embedding of size 400. For master gates, the downsize factor $C = 10$. The total number of parameters was slightly increased from 24 millions to 25 millions with additional matrices for computing master gates. We manually searched some of the dropout values for ON-LSTM based on the validation performance. The values used for dropout on the word vectors, the output between LSTM layers, the output of the final LSTM layer, and embedding dropout where (0.5, 0.3, 0.45, 0.1) respectively. A weight-dropout of 0.45 was applied to the recurrent weight matrices.

As shown in Table 1, our model performs better than the standard LSTM while sharing the same number of layers, embedding dimensions, and hidden states units. Recall that the master gates only control how information is stored in different neurons. It is interesting to note that we can improve

the performance of a strong LSTM model without adding skip connections or a significant increase in the number of parameters.

## 5.2 UNSUPERVISED CONSTITUENCY PARSING

The unsupervised constituency parsing task compares the latent stree structure induced by the model with those annotated by human experts. Following the experiment settings proposed in Htut et al. (2018), we take our best model for the language modeling task, and test it on WSJ10 dataset and WSJ test set. WSJ10 has 7422 sentences, filtered from the WSJ dataset with the constraint of 10 words or less, after the removal of punctuation and null elements (Klein & Manning, 2002). The WSJ test set contains 2416 sentences with various lengths. It is worth noting that the WSJ10 test set contains sentences from the training, validation, and test set of the PTB dataset, while WSJ test uses the same set of sentences as the PTB test set.

To infer the tree structure of a sentence from a pre-trained model, we initialize the hidden states with the zero vector, then feed the sentence into the model as done in the language modeling task. At each time step, we compute an estimate of $d_t^f$:

$$\hat{d}_t^f = \mathbb{E}\left[d_t^f\right] = \sum_{k=1}^{D_m} k p_f(d_t = k) = \sum_{k=1}^{D_m} \sum_{i=1}^{k} p_f(d_t = k) = D_m - \sum_{k=1}^{D_m} \tilde{f}_{tk} \tag{15}$$

where $p_f$ is the probability distribution over split points associated to the master forget gate and $D_m$ is the size of the hidden state. Given $\hat{d}_t^f$, we can use the top-down greedy parsing algorithm proposed in Shen et al. (2017) for unsupervised constituency parsing. We first sort the $\{\hat{d}_t^f\}$ in decreasing order. For the first $\hat{d}_i^f$ in the sorted sequence, we split the sentence into constituents $((x_{<i}), (x_i, (x_{>i})))$. Then, we recursively repeat this operation for constituents $(x_{<i})$ and $(x_{>i})$, until each constituent contains only one word.

The performance is shown in Table 2. The second layer of ON-LSTM achieves state-of-the-art unsupervised constituency parsing results on the WSJ test set, while the first and third layers do not perform as well. One possible interpretation is that the first and last layers may be too focused on capturing local information useful for the language modeling task as they are directly exposed to input tokens and output predictions respectively, thus may not be encouraged to learn the more abstract tree structure. Since the WSJ test set contains sentences of various lengths which are unobserved during training, we find that ON-LSTM provides better generalization and robustness toward longer sentences than previous models. We also see that ON-LSTM model can provide strong results for phrase detection, including ADJP (adjective phrases), PP (prepositional phrases), and NP (noun phrases). This feature could benefit many downstream tasks, like question answering, named entity recognition, co-reference resolution, etc.

## 5.3 TARGETED SYNTACTIC EVALUATION

Targeted syntactic evaluation tasks have been proposed in Marvin & Linzen (2018). It is a collection of tasks that evaluate language models along three different structure-sensitive linguistic phenomena: subject-verb agreement, reflexive anaphora and negative polarity items. Given a large number of minimally different pairs of English sentences, each consisting of a grammatical and an ungrammatical sentence, a language model should assign a higher probability to a grammatical sentence than an ungrammatical one.

Using the released codebase[2] and the same settings proposed in Marvin & Linzen (2018), we train both our ON-LSTM model and a baseline LSTM language model on a 90 million word subset of Wikipedia. Both language models have two layers of 650 units, a batch size of 128, a dropout rate of 0.2, a learning rate of 20.0, and were trained for 40 epochs. The input embeddings have 200 dimensions and the output embeddings have 650 dimesions.

Table 3 shows that the ON-LSTM performs better on the long-term dependency cases, while the baseline LSTM fares better on the short-term ones. This is possibly due to the relatively small num-

---

[2] https://github.com/BeckyMarvin/LM_syneval. We notice that the test set generated from the code is different from the one used in the original paper Marvin & Linzen (2018). Therefore, our results are not strictly comparable with the results in Marvin & Linzen (2018).

| Model | Training Data | Training Object | Vocab Size | Parsing F1 | | | | Depth WSJ | Accuracy on WSJ by Tag | | | |
| | | | | WSJ10 $\mu$ ($\sigma$) | max | WSJ $\mu$ ($\sigma$) | max | | ADJP | NP | PP | INTJ |
|---|---|---|---|---|---|---|---|---|---|---|---|---|
| PRPN-UP | AllNLI Train | LM | 76k | 66.3 (0.8) | 68.5 | 38.3 (0.5) | 39.8 | 5.8 | 28.7 | 65.5 | 32.7 | *0.0* |
| PRPN-LM | AllNLI Train | LM | 76k | 52.4 (4.9) | 58.1 | 35.0 (5.4) | 42.8 | 6.1 | 37.8 | 59.7 | **61.5** | **100.0** |
| PRPN-UP | WSJ Train | LM | 15.8k | 62.2 (3.9) | 70.3 | 26.0 (2.3) | 32.8 | 5.8 | 24.8 | 54.4 | 17.8 | *0.0* |
| PRPN-LM | WSJ Train | LM | 10k | 70.5 (0.4) | 71.3 | 37.4 (0.3) | 38.1 | 5.9 | 26.2 | **63.9** | 24.4 | *0.0* |
| **ON-LSTM** 1st-layer | WSJ Train | LM | 10k | 35.2 (4.1) | 42.8 | 20.0 (2.8) | 24.0 | 5.6 | 38.1 | 23.8 | 18.3 | **100.0** |
| **ON-LSTM** 2nd-layer | WSJ Train | LM | 10k | 65.1 (1.7) | 66.8 | 47.7 (1.5) | **49.4** | 5.6 | **46.2** | 61.4 | 55.4 | *0.0* |
| **ON-LSTM** 3rd-layer | WSJ Train | LM | 10k | 54.0 (3.9) | 57.6 | 36.6 (3.3) | 40.4 | 5.3 | 44.8 | 57.5 | 47.2 | *0.0* |
| 300D ST-Gumbel | AllNLI Train | NLI | – | – | – | 19.0 (1.0) | 20.1 | – | *15.6* | *18.8* | *9.9* | 59.4 |
| w/o Leaf GRU | AllNLI Train | NLI | – | – | – | 22.8 (1.6) | 25.0 | – | 18.9 | 24.1 | *14.2* | 51.8 |
| 300D RL-SPINN | AllNLI Train | NLI | – | – | – | *13.2 (0.0)* | 13.2 | – | *1.7* | *10.8* | *4.6* | 50.6 |
| w/o Leaf GRU | AllNLI Train | NLI | – | – | – | *13.1 (0.1)* | 13.2 | – | *1.6* | *10.9* | *4.6* | 50.0 |
| CCM | WSJ10 Full | – | – | – | 71.9 | – | – | – | – | – | – | – |
| DMV+CCM | WSJ10 Full | – | – | – | 77.6 | – | – | – | – | – | – | – |
| UML-DOP | WSJ10 Full | – | – | – | **82.9** | – | – | – | – | – | – | – |
| Random Trees | – | – | – | 31.7 (0.3) | 32.2 | 18.4 (0.1) | 18.6 | 5.3 | 17.4 | 22.3 | 16.0 | 40.4 |
| Balanced Trees | – | – | – | 43.4 (0.0) | 43.4 | 24.5 (0.0) | 24.5 | 4.6 | 22.1 | *20.2* | *9.3* | 55.9 |
| Left Branching | – | – | – | *19.6 (0.0)* | *19.6* | *9.0 (0.0)* | *9.0* | 12.4 | – | – | – | – |
| Right Branching | – | – | – | 56.6 (0.0) | 56.6 | 39.8 (0.0) | 39.8 | 12.4 | – | – | – | – |

Table 2: Unlabeled parsing F1 results evaluated on the full WSJ10 and WSJ test set. Our language model has three layers, each of them provides a sequence of $\hat{d}_t^f$. We provide the parsing performance for all layers. Results with RL-SPINN and ST-Gumbel are evaluated on the full WSJ (Williams et al., 2017). PRPN models are evaluated on the WSJ test set (Htut et al., 2018). We run the model with 5 different random seeds to calculate the average F1. The *Accuracy* columns represent the fraction of ground truth constituents of a given type that correspond to constituents in the model parses. We use the model with the best F1 score to report ADJP, NP, PP, and INTJ. WSJ10 baselines are from Klein & Manning (2002, CCM), Klein & Manning (2005, DMV+CCM), and Bod (2006, UML-DOP). As the WSJ10 baselines are trained using POS tags, they are not strictly comparable with the latent tree learning results. Italics mark results that are worse than the random baseline.

ber of units in the hidden states, which is insufficient to take into account both long and short-term information. We also notice that the results for NPI test cases have unusually high variance across different hyper-parameters. This result maybe due to the non-syntactic cues discussed in Marvin & Linzen (2018). Despite this, ON-LSTM actually achieves better perplexity on the validation set.

## 5.4 LOGICAL INFERENCE

We also analyze the model's performance on the logical inference task described in Bowman et al. (2015). This task is based on a language that has a vocabulary of six words and three logical operations, $or, and, not$. There are seven mutually exclusive logical relations that describe the relationship between two sentences: two types of entailment, equivalence, exhaustive and non-exhaustive contradiction, and two types of semantic independence. Similar to the natural language inference task, this logical inference task requires the model to predict the correct label given a pair of sentences. The train/test split is as described in the original codebase[3], and 10% of training set is set aside as the validation set.

We evaluate the ON-LSTM and the standard LSTM on this dataset. Given a pair of sentences $(s_1, s_2)$, we feed both sentences into an RNN encoder, taking the last hidden state $(h_1, h_2)$ as the sentence embedding. The concatenation of $(h_1, h_2, h_1 \circ h_2, \text{abs}(h_1 - h_2))$ is used as input to a multi-layer classifier, which gives a probability distribution over seven labels. In our experiment, the RNN models were parameterised with 400 units in one hidden layer, and the input embedding size was 128. A dropout of 0.2 was applied between different layers. Both models are trained on sequences with 6 or less logical operations and tested on sequences with at most 12 operations.

Figure 3 shows the performance of ON-LSTM and standard LSTM on the logical inference task. While both models achieve nearly 100% accuracy on short sequences ($\leq 3$), ON-LSTM attains

---

[3] `https://github.com/sleepinyourhat/vector-entailment`

|  | ON-LSTM | LSTM |
|---|---|---|
| **Short-Term Dependency** | | |
| SUBJECT-VERB AGREEMENT: | | |
| Simple | 0.99 | **1.00** |
| In a sentential complement | 0.95 | **0.98** |
| Short VP coordination | 0.89 | **0.92** |
| In an object relative clause | 0.84 | **0.88** |
| In an object relative (no *that*) | 0.78 | **0.81** |
| REFLEXIVE ANAPHORA: | | |
| Simple | **0.89** | 0.82 |
| In a sentential complement | **0.86** | 0.80 |
| NEGATIVE POLARITY ITEMS: | | |
| Simple (grammatical vs. intrusive) | 0.18 | **1.00** |
| Simple (intrusive vs. ungrammatical) | **0.50** | 0.01 |
| Simple (grammatical vs. ungrammatical) | 0.07 | **0.63** |
| **Long-Term Dependency** | | |
| SUBJECT-VERB AGREEMENT: | | |
| Long VP coordination | **0.74** | **0.74** |
| Across a prepositional phrase | 0.67 | **0.68** |
| Across a subject relative clause | **0.66** | 0.60 |
| Across an object relative clause | **0.57** | 0.52 |
| Across an object relative (no *that*) | **0.54** | 0.51 |
| REFLEXIVE ANAPHORA: | | |
| Across a relative clause | 0.57 | **0.58** |
| NEGATIVE POLARITY ITEMS: | | |
| Across a relative clause (grammatical vs. intrusive) | 0.59 | **0.95** |
| Across a relative clause (intrusive vs. ungrammatical) | **0.20** | 0.00 |
| Across a relative clause (grammatical vs. ungrammatical) | **0.11** | 0.04 |

Table 3: Overall accuracy for the ON-LSTM and LSTM on each test case. "Long-term dependency" means that an unrelated phrase (or a clause) exist between the targeted pair of words, while "short-term dependency" means there is no such distraction.

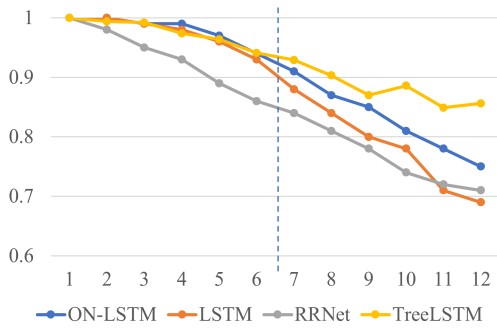

Figure 3: Test accuracy of the models, trained on short sequences ($\leq 6$) in logic data. The horizontal axis indicates the length of the sequence, and the vertical axis indicates the accuracy of models performance on the corresponding test set.

better performance on sequences longer then 3. The performance gap continues to increase on longer sequences ($\geq 7$) that were not present during training. Hence, the ON-LSTM model shows better generalization while facing structured data with various lengths and comparing to the standard LSTM. A tree-structured model can achieve strong performance on this dataset (Bowman et al., 2015), since it is provided with the ground truth structure as input. The recursive application of the same composition function is well suited for this task. We also include the result of RRNet (Jacob et al., 2018), which can induce the latent tree structure from downstream tasks. Note that the results may not be comparable, because the hyper-parameters for training were not provided.

## 6 CONCLUSION

In this paper, we propose *ordered neurons*, a novel inductive bias for recurrent neural networks. Based on this idea, we propose a novel recurrent unit, the ON-LSTM, which includes a new gating mechanism and a new activation function $\mathrm{cumax}(\cdot)$. This brings recurrent neural networks closer to performing tree-like composition operations, by separately allocating hidden state neurons with long and short-term information. The model performance on unsupervised constituency parsing shows that the ON-LSTM induces the latent structure of natural language in a way that is coherent with human expert annotation. The inductive bias also enables ON-LSTM to achieve good performance on language modeling, long-term dependency, and logical inference tasks.

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

## A  SAMPLE PARSES FROM THE MODEL WITH THE BEST PERPLEXITY

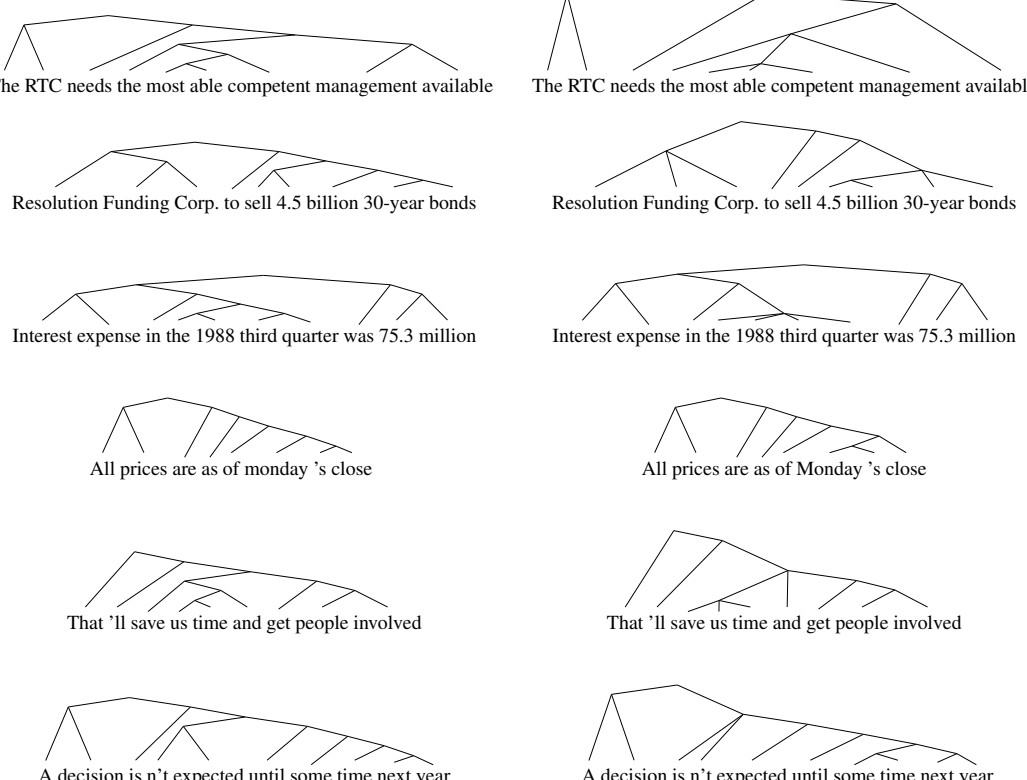

Figure A.1: *Left* parses are from the 2nd layer of the ON-LSTM model, *Right* parses are converted from human expert annotations (removing all punctuations).

