# OpenReview forum: "Ordered Neurons: Integrating Tree Structures into Recurrent Neural Networks"
_ICLR.cc/2019/Conference_

### Official Review · AnonReviewer3 · 2018-10-21
**Promising approach for adding a hierarchical inductive bias to LSTMs**

**Rating:** 8
**Confidence:** 4

**Review:**

Language is hierarchically structured: smaller units (e.g., noun phrases) are nested within larger units (e.g., clauses). This is a strict hierarchy: when a larger constituent ends, all of the smaller constituents that are nested within it must also be closed. While the different units of an LSTM can learn to track information at different time scales, the standard architecture does not impose this sort of strict hierarchy. This paper proposes to add this constraint to the system by ordering the units; a vector of "master" input and forget gates ensures that when a given unit is reset all of the units that follow it in the ordering are also reset.

Strengths:
* The paper introduces an elegant way of adding a hierarchical inductive bias; the intuition behind this idea is explained clearly.
* The evaluation tasks are very sensible. It's good that the model is shown to obtain good perplexity and slightly improve over an LSTM baseline; it's not the state of the art, but that's not the point of the paper (in fact, I would emphasize that even more than the authors do). The unsupervised parse evaluation (Table 2) is the heart of the paper, in my opinion (and should probably be emphasized more) -- the results from the second layer are quite impressive.
* The (mildly) better performance than LSTMs on long-distance dependencies, and (mildly) worse performance on local dependencies, in the Marvin & Linzen dataset, is interesting (and merits additional analysis).

Weaknesses:
* The discussion of the motivation for unsupervised structure induction in the introduction is somewhat confused. I am not sure that neural networks with latent syntactic structures can really address the seemingly very fundamental question mentioned in the first paragraph (whether syntax is related to "an underlying mechanism of human cognition") - I would suggest eliminating this part. At the same time, the authors might want to add another motivation for studying architectures that discover latent structure (as opposed to being given that structure) - this setting corresponds more closely to human language acquisition, where children aren't given annotated parse trees.
* The authors discuss hierarchy in terms of syntactic structure alone, but it would seem to me that the hierarchy that the LSTM is inducing could just as well include topic shifts, speech acts and others, especially if the network is trained across sentences.
* There is limited analysis of the model. Why does the second layer show better unsupervised parsing performance than the third layer? (Could this be related to syntactic vs. semantic/discourse units I mention in the previous bullet?) Why is the model better at ADJP boundaries than NP boundaries? It would have been more useful to report less experiments but analyze the results of each experiment in greater depth.
* In this vein, I am not sure it's useful to include WSJ10 in Table 2, which is busy as it is. These sentences are clearly too easy, as the right branching baseline shows, and require additional POS tagging.
* I found it difficult to read Figure A.2: could you help us understand what we should take away from it?
* It is not entirely clear why the model needs both unit-specific forget/input gates and the "master" forget/input gates, and there is no discussion of this issue. Have you tried using only the "master" gates?

Minor notes:
* RNNGs are described as having an explicit bias to model syntactic structure; this is an arguably confusing use of the word "bias", in that the architecture has a hard constraint enforcing syntactic structures (bias implies a soft constraint).
* There are some language issues: agreement errors (e.g. "have" in the sentence that starts with "Developing" in the introduction), typos ("A order should exist", "co-occurance"), determiner issues ("values in [the] master forget gate", "when the overlap exists") - I would suggest going through and copy editing the paper.
* "cummax" seems like a better choice of name for cumulative maximum than "cumax".
* It may be helpful to remind the reader of the update equation for c_t in a standard LSTM.
* Did the language model have 1150 units in each layer or in total? Why did you use exactly three layers? Did you try one, two and four?
* It's not clear if the results in Table 2 reflect the best seed out of five (as the title of the column "max" indicates) or the average (as the caption says).

---

> ### Author Response · Authors · 2018-11-19
> **Thank you for the review and comments. We have made some modifications to the paper based on some of your feedback.**
>
> Regarding “The discussion of the motivation for unsupervised structure induction in the introduction is somewhat confused”.
> Thanks for this suggestion. We have modified our introduction accordingly.
>
> Regarding “the author discuss hierarchy in terms of syntactic structure alone”.
> It’s possible that the learned hierarchy also reflects topic and other structures. However, to our knowledge, it maybe hard to quantitatively measure whether the model captures such semantic level structures. We will further study the relationship between the induced structure and semantic units.
>
> Regarding “Why does the second layer show better unsupervised parsing performance than the third layer?”.
> Our hypothesis is that the first and last layer focus on low level or short term information, while the middle also include longer term information. Similar results can be found in [1].
> [1]Blevins, Terra, Omer Levy, and Luke Zettlemoyer. "Deep RNNs Encode Soft Hierarchical Syntax." arXiv preprint arXiv:1805.04218 (2018).
>
> Regarding “Have you tried using only the "master" gates?“
> We did try this. We found that for better language modelling performance, we still needed to use the unit specific gates. The unsupervised parsing capability of a master-gate-only model was similar.
>
> Regarding “Did the language model have 1150 units in each layer or in total? Why did you use exactly three layers? Did you try one, two and four?”
> The main reason for these choices was to compare with the AWD-LSTM model, so we followed the hyperparameters used there as closely as possible. The first and second layer uses 1150 units, the last layer has 400 units. Having tried the one and two layer settings, we find that the hyperparameters do not result in similar parsing performance.
>
>
> Regarding “It's not clear if the results in Table 2 reflect the best seed out of five”
> The “max” column provides the best result among different random seeds, while the “µ (σ)” column provides the mean and variance.

---

### Official Review · AnonReviewer2 · 2018-10-28
**Solid contribution**

**Rating:** 7
**Confidence:** 3

**Review:**

The paper proposes a new RNN unit: ON-LSTM. The idea is to explicitly integrates the latent tree structure into recurrent models. Experiments are conducted to evaluate performances on four different tasks: language modeling, unsupervised parsing, targeted syntactic evaluation, and logical inference. Good results on unsupervised parsing show that the model learns something close to human judgments of the sentence parses.

The paper is clearly written, and the experiments seem planned well.
The language modeling results are not state-of-the-art, but the unsupervised parsing results of layer 2 are quite impressive. The analyses are reasonable.

Overall, the paper seems worthy of being accepted.

---

> ### Author Response · Authors · 2018-11-19
> **Thank you for the review and comments.**
>
> It is true that our language modeling results are not state-of-the-art. The ordered neurons primarily focuses on inducing the latent structure of sequential data. We wanted to demonstrate that the model is able to give good parsing results in the acceptable range in terms of perplexity. One of the future research direction is trying to improve current SOTA models with the ordered neurons.

---

> ### Comment · AnonReviewer2 · 2018-11-25
> **Some additional notes**
>
> Overall, it seems that all reviewers agree that this paper is quite interesting. However, since my original review was rather short, I wanted to give some additional notes/suggestions:
> - In the abstract you say that the LSTM's "performance consistently lags behind that of tree-based models". Is that actually true? If so, could you maybe give examples for tasks where this is the case? (Otherwise, maybe weaken that statement.)
> - I felt that the explanation for the performance difference between the different layers for unsupervised parsing is rather vague, given that this is arguably the most important part of the paper. You mentioned earlier work which found something similar in another response and said that parsing wasn't as good for one or two layers. However, there are some additional interesting questions to ask, e.g., what happens with 4 layers? And is there a performance difference for parsing between the two layers when using two?
> - Concerning the logical inference experiment: Is there a reason why you train on <=6 operations only, but evaluate on longer sequences?
> - Finally, I agree with the other reviewers that the paper needs some editing. More examples are, e.g., Section 5.2.: "latent stree structure" -> "latent tree structures"; Section 5.3.: "ON-LSTM perform" -> "ON-LSTM performs"; Section 5.4.: "given pair of sentences" -> "given pairs of sentences", etc.

---

> > ### Author Response · Authors · 2018-11-26
> > **Thanks for your additional notes. We have made some modifications to the paper based on some of your feedback.**
> >
> > Regarding “LSTM’s performance consistently lags behind that of tree-based models”.
> > On sentence embedding tasks (e.g SNLI) and sequential labeling tasks (e.g sentiment analysis), TreeLSTM has shown better performance compared to vanilla LSTM. We’ve also updated the abstract according to the reviews.
> >
> > Regarding “the performance difference for different layers”.
> > Thanks for the suggestion. The main reason for choosing the number of layers was to compare with the AWD-LSTM model, so we followed the hyperparameters used there as closely as possible. We will further investigate the relationship between varying the number of layers and its effect on the parsing performance. Our hypothesis is that the first and last layer focus on low level or short term information, while the middle also include longer term information. Similar results can be found in [1].
> > [1]Blevins, Terra, Omer Levy, and Luke Zettlemoyer. "Deep RNNs Encode Soft Hierarchical Syntax." arXiv preprint arXiv:1805.04218 (2018).
> >
> > Regarding the “logical inference experiment”.
> > We follow the same experiment setting in previous paper. It's designed to test the generalizability of model.

---

### Official Review · AnonReviewer1 · 2018-11-01
**A very interesting new proposal, thoroughly explored, with at least middling good results**

**Rating:** 9
**Confidence:** 4

**Review:**

Quality
 - Pro:
   o This paper was in general a quality effort. It had a thorough bibliography of both older and recent relevant research contributions
   o Providing useful, well done experimental results on four tasks was also a sign of this good thoroughness
 - Con: none observed

Clarity
 - Pro:
   o The paper was generally well-written and clear. Results were clearly presented.
 - Con:
   o Notwithstanding the half page of explanation of the intuition behind the new ON-LSTM update rules (top of p.5), it wasn't really enough for my old brain to get a good sense of what was going on – though I'm sure younger, smarter people will have made more sense of it. :) It would really help to try to provide more intuition and understanding here. Things that would probably really help include a worked example and diagrams.
   o There were minor English/copyediting problems, but nothing that interfered with understanding. E.g., "monotonously" on p.4 should be "monotonically" (twice).

Originality
 - Pro
   o This was REALLY NEAT! This paper had a real, clear, different idea that appeared interesting and promising. That puts it into the top half of accepted papers right there.
   o The basic idea of the different update time scales, done flexibly, controlled by the master forget/input gates seemed original, flexible, and good.
 - Con: Nothing really observed; there are clearly a bunch of slightly related ideas, well referenced in this paper.

Significance
 - Pro
   o If this idea pans out well, it would be a really interesting new structural prior to add to the somewhat impoverished vocabulary of successful techniques for building deep learning systems.
   o Has an original, promising approach. That has the opportunity for impact and significance.
 - Con:
   o The results so far are interesting, and in places promising, but not so clearly good that this idea doesn't need further evaluation of its usefulness.
   o All the results presented are on small datasets (Penn Treebank WSJ (1 million words) size or smaller). What are the prospects on bigger datasets?  It looks like in principle this shouldn't be a big obstacle – except for not having a highly tuned CuDNN implementation, it looks like this should basically be fairly efficient like an LSTM and not hard to scale like, e.g., an RNNG.

Other comments:
 - Some of the wording on page 1 seemed strange to me. Natural language has a linear overt form as spoken and (hence) written. It's really not that the sequential form is just how people conventionally "present" it. That is, it's not akin to a chemical compound which is really 3 dimensional but commonly "presented" by chemists in a convenient sequential notation.
 - p.2 2nd paragraph: Don't RNNs "explicitly impose a chain structure" not "implicitly"?!?
 - I wasn't sure I was sold on the name "Ordered Neurons". I'm not sure I have the perfect answer here, but it feels more like "multi-timescale units" is what is going on.
 - The LM results look good.
 - Because of all the different datasets, etc. it was a little hard to call the grammar induction results, but they at least look competently strong.
 - The stronger results on long dependencies in targeted syntactic evaluation look promising, but maybe you need a bigger hidden size so you can also do as well on short dependencies?
 - The logical inference results were promising – they seem to suggest that you capture some but not all of the value of explicit tree structure (a TreeLSTM) on a task like this.
 - The tree structures in Appendix A look promisingly good.

---

> ### Author Response · Authors · 2018-11-19
> **Thanks for your review and kind comments. We have made some modifications to the paper based on some of your feedbacks.**
>
> Regarding the “hidden size” for targeted syntactic evaluation.
> For fair comparison, we kept the number of parameters of our model comparable with the baseline. We also tried bigger hidden sizes and more layers. We observe that increasing the capacity doesn’t improve the performance on the task. This maybe due to overfitting. Since the training set and test set don’t share the same data distribution, overfitting the training set doesn’t necessarily provide better results on the test set.
>
> Regarding the “logical inference results”.
> Understanding the causes of that gap and investigating to what extent we can fill that gap is an important future direction. The advantages of the TreeLSTM over the ON-LSTM are 1) TreeLSTMs have access to the true structure; 2) TreeLSTMs reuse weights across the compositional processes related to non-terminal nodes, while the ON-LSTMs don’t have shared weights across different levels in the tree. For this task, the test set contains data with deeper tree structures than those seen during training. The weight sharing feature in TreeLSTMs may be beneficial in order to achieve better generalization.
>
> Regarding the wording in the introduction.
> Thank you for the valuable suggestions. We have reworded the introduction to make it clear that the overtly sequential form is an essential characteristic for natural language, not just a conventional presentation format. In addition, we have changed the sentence to say that RNN explicitly imposes a chain structure.

---

> > ### Comment · AnonReviewer1 · 2018-11-22
> > **Thanks!**
> >
> > Small changes look good to me.

---

### Meta-Review · Area_Chair1 · 2018-12-13
**Original idea, substantial experiments to back it up**

**Confidence:** 4
**Recommendation:** Accept (Oral)

**Metareview:**

This paper presents a substantially new way of introducing a syntax-oriented inductive bias into sentence-level models for NLP without explicitly injecting linguistic knowledge. This is a major topic of research in representation learning for NLP, so to see something genuinely original work well is significant. All three reviewers were impressed by the breadth of the experiments and by the results, and this will clearly be among the more ambitious papers presented at this conference.

In preparing a final version of this paper, though, I'd urge the authors to put serious further effort into the writing and presentation. All three reviewers had concerns about confusing or misleading passages, including the title and the discussion of the performance of tree-structured models so far.